# Deep LiDAR-Radar-Visual Fusion for Object Detection in Urban Environments

Yuhan Xiao [1,†], Yufei Liu [1,†], Kai Luan [1], Yuwei Cheng [2,3], Xieyuanli Chen [1] and Huimin Lu [1,*]

1   College of Intelligence Science and Technology, National University of Defense Technology, No. 137 Yanwachi Street, Changsha 410073, China; xiaoyuhan20@nudt.edu.cn (Y.X.); yufei.liu@nudt.edu.cn (Y.L.); luankai0504@foxmail.com (K.L.); xieyuanli.chen@nudt.edu.cn (X.C.)
2   Department of Electronic Engineering, Tsinghua University, Beijing 100084, China; chengyw18@mails.tsinghua.edu.cn
3   ORCA-TECH, 9–11 Myrtle St., North Sydney, NSW 2060, Australia
*   Correspondence: lhmnew@nudt.edu.cn
†   These authors contributed equally to this work.

**Abstract:** Robust environmental sensing and accurate object detection are crucial in enabling autonomous driving in urban environments. To achieve this goal, autonomous mobile systems commonly integrate multiple sensor modalities onboard, aiming to enhance accuracy and robustness. In this article, we focus on achieving accurate 2D object detection in urban autonomous driving scenarios. Considering the occlusion issues of using a single sensor from a single viewpoint, as well as the limitations of current vision-based approaches in bad weather conditions, we propose a novel multi-modal sensor fusion network called LRVFNet. This network effectively combines data from LiDAR, mmWave radar, and visual sensors through a deep multi-scale attention-based architecture. LRVFNet comprises three modules: a backbone responsible for generating distinct features from various sensor modalities, a feature fusion module utilizing the attention mechanism to fuse multi-modal features, and a pyramid module for object reasoning at different scales. By effectively fusing complementary information from multi-modal sensory data, LRVFNet enhances accuracy and robustness in 2D object detection. Extensive evaluations have been conducted on the public VOD dataset and the Flow dataset. The experimental results demonstrate the superior performance of our proposed LRVFNet compared to state-of-the-art baseline methods.

**Keywords:** multi-sensor fusion; multi-modal sensing; object detection; deep learning method

## 1. Introduction

Sensing the environment entails utilizing diverse sensors to acquire detailed environmental measurements and accurately identify objects of interest, which is pivotal for versatile applications, including autonomous driving, robotics, and computer vision. Object detection is one of the most important perception tasks, serving as the foundation for mobile autonomous systems to conduct subsequent tasks. Various sensor types, such as cameras, LiDAR, and radar, have been applied for mobile system perception. The importance of using multiple sensors for perception lies in their complementary capabilities. Each type of sensor has its strengths and weaknesses in detecting different objects under different environmental conditions.

Cameras are one of the most frequently used sensors providing high-resolution images. However, the stability and reliability of camera detection are limited due to lighting, shadows, and dynamic changes. Therefore, cameras are mainly utilized for image semantic segmentation and 2D object detection. LiDAR sensors scan the surrounding environment and provide accurate 3D information. With its precise range of measurements, LiDAR has a natural advantage in object detection. However, LiDAR is significantly affected by various weather conditions, such as rain, fog, and sandstorms, impacting its measurement

accuracy. Recently, millimeter-wave radar (mmWave radar) has been used for measuring the location and velocity of objects. Its low working frequency enables penetration through thick clouds, fog, rain, and snow, facilitating object detection in diverse weather conditions. However, mmWave radar exhibits lower resolution and lacks precise 3D data capability.

Utilizing multiple sensor modalities has the potential to enhance the reliability and precision of sensing tasks. However, it also presents new challenges in designing the perception system. Early fusion approaches [1] typically combine raw or pre-processed sensor data from different modalities, but they are often susceptible to spatial or temporal misalignment. Conversely, late fusion approaches [2,3] integrate data from various modalities at the decision level, offering greater flexibility to incorporate novel sensing modalities into the network. However, late fusion approaches fail to fully exploit the potential of available sensing modalities as they do not leverage the intermediate features acquired through joint representation learning. Moreover, existing sensor fusion methods primarily combine camera data with either LiDAR [4,5] or mmWave radar data [6–8]. The effective integration of data from vision, LiDAR, and mmWave radar sensors to achieve accurate object detection in multimodal fusion has been inadequately explored.

In this article, we present a novel approach called Deep LiDAR-Radar-Visual Fusion Network (LRVFNet) for robust and precise object detection in complex urban driving scenarios. Our LRVFNet leverages the strengths of multiple sensors, including LiDAR, mmWave radar, and vision, as shown in Figure 1, to effectively fuse their respective data and extract meaningful features. Our LRVFNet aims to address the challenges posed by occlusions, cluttered scenes, and varying lighting conditions in urban environments using spatial–temporal multimodal data. Through the proposed feature-level fusion module incorporating self-attention and global attention mechanisms, the LRVFNet seamlessly integrates the extracted features from visual and BEV images generated by LiDAR and mmWave radar data. These features are further refined using a convolutional layer to construct an efficient feature pyramid network (FPN). By exploiting the multi-scale information obtained from the FPN, our proposed method significantly improves the detection performance for heavily occluded objects and small targets. Extensive evaluations have been conducted on two public datasets, the VoD dataset [9] and the Flow dataset [10]. The experimental results demonstrate the superiority of our method compared to the state-of-the-art baseline methods on object detection in urban driving environments.

The main contributions of this paper are threefold:

- A novel framework LRVFNet is proposed to achieve accurate 2D object detection in urban environments exploiting spatial–temporal LiDAR, mmWave radar, and vision.
- A novel feature-level fusion module is proposed, which effectively integrates LiDAR and mmWave radar features into 2D object detection using self-attention and global attention mechanisms, showing significantly improved fusion performance compared to existing fusion schemes.
- Extensive experimental results on two public VoD and Flow datasets demonstrate that our LRVFNet outperforms the state-of-the-art model with large margins by 6.43% and 2.61% in terms of 2D object detection accuracy $AP^{50}$. Additionally, the ablation study validates the effectiveness of our network designs.

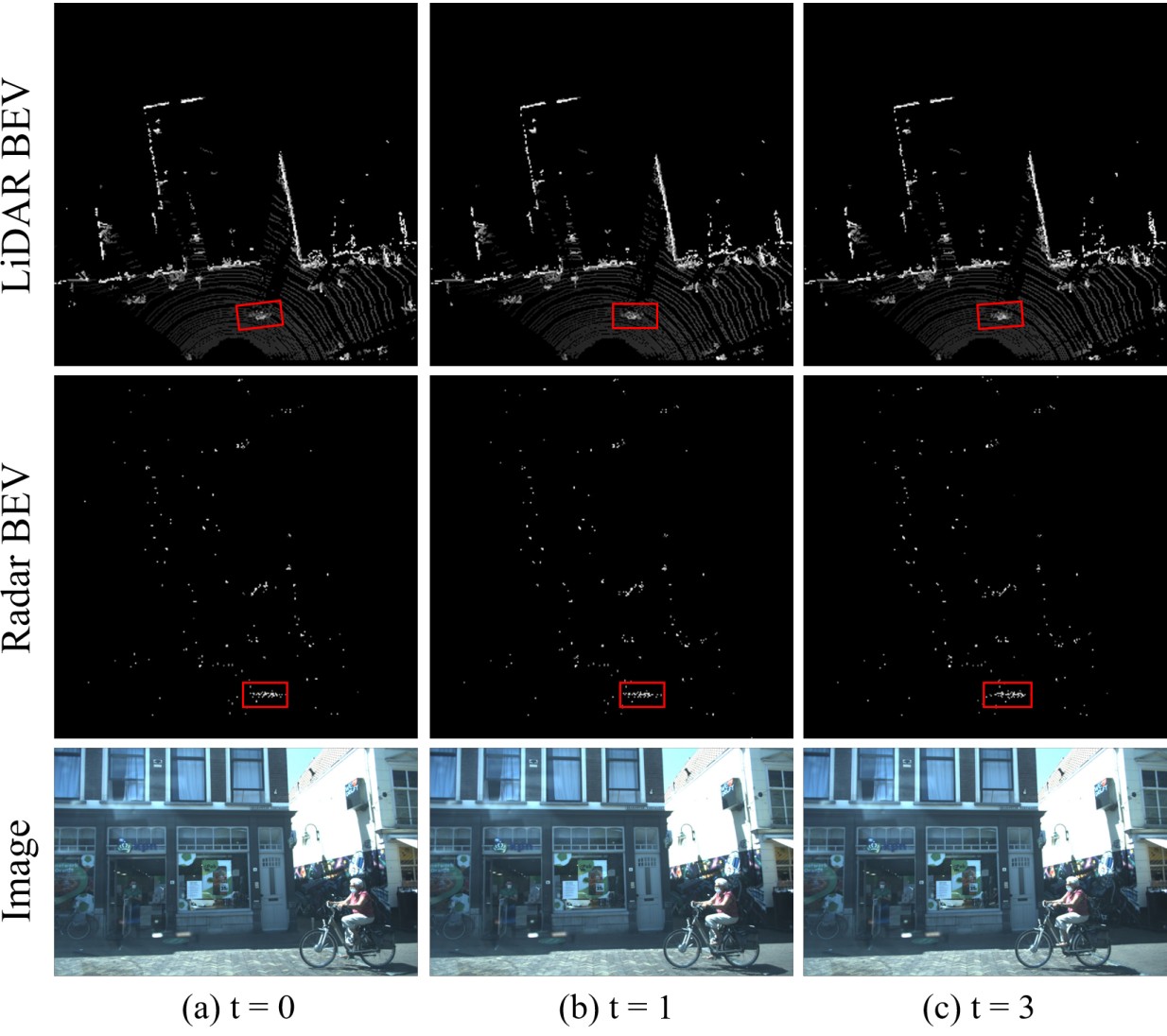

**(a) t = 0**        **(b) t = 1**        **(c) t = 3**

**Figure 1.** The outcome of converting the point cloud from LiDAR, mmWave radar into bird's-eye view (BEV) is depicted. The red box highlights a pedestrian riding a bicycle, clearly illustrating the movement process. Compared to LiDAR, the mmWave radar is significantly sparser, resulting in limited available information.

## 2. Materials

### 2.1. Single-Modality Methods

Numerous studies have been conducted on object detection using various single sensory modalities, such as cameras, LiDAR, and mmWave radar sensors. Regarding visual-based approaches, the integration of segmentation algorithms into AlexNet [11] for region proposal sampling was pioneered by R-CNN [12]. Faster R-CNN [13] improves the integration of a region proposal network (RPN) and Fast R-CNN [14] by introducing attention mechanisms that facilitate the sharing of convolutional features within a unified network. Tang et al. [15] introduced a weighted bidirectional feature network and a customized compound scaling method to enhance both accuracy and efficiency. Additionally, Redmon et al. [16] devised the You Only Look Once (YOLO) sequential approaches, achieving outstanding performance in 2D object detection. YOLOv5 [17] is a real-time object detection algorithm that presents a unique method for optimizing anchor boxes and focuses on improving small object detection. YOLOX [18] builds upon YOLOv5 [17] and introduces an improved anchor-free mechanism for object detection. YOLOR [19] efficiently incorporates both explicit and implicit knowledge, acquired through subcon-

scious learning, to enable effective multi-task learning within a unified model architecture. The groundbreaking object detection algorithm YOLOv7-E6 [20] introduced an innovative real-time architecture for efficient object detection, along with a corresponding model scaling technique. This method has demonstrated exceptional progress in terms of both speed and accuracy. In addition to the YOLO series, DETR [21] is a transformer-based object detection algorithm that revolutionizes the conventional anchor-based approach by employing a set prediction paradigm. It has achieved state-of-the-art performance and showcased the potential of transformers in computer vision tasks. Deformable DETR [22] extends the functionality of the DETR algorithm by integrating deformable convolution and deformable attention modules, thereby enhancing the resilience of object detection when faced with deformations and occlusions.

Compared to methods based on visual data, there has been relatively less research conducted on LiDAR- and radar-based methods. Jin et al. [23] proposed a robust vehicle detection approach based on LiDAR, consisting of three components: point cloud clustering, bounding box fitting, and point cloud recognition. To eliminate the need for manual feature engineering on 3D point clouds, Zhou et al. [24] introduced VoxelNet, a universal 3D detection network that integrates feature extraction and bounding box prediction into a single-step, end-to-end trainable deep network. Zeng et al. [25] presented a real-time three-dimensional (RT3D) vehicle detection method that utilizes pure LiDAR point clouds to predict the location, orientation, and size of vehicles. Svenningsson et al. [26] proposed an object recognition model that imposes a graph structure on the mmWave radar point cloud by connecting spatially proximal points. The model extracts local patterns by performing convolutional operations across the graph's edges. On the other hand, Meyer et al. [27] focused on fully leveraging raw mmWave radar tensor data instead of relying on human-biased point clouds, which are typically generated using traditional mmWave radar signal processing techniques. They presented a network for 3D object detection solely based on mmWave radar data. Their object detection method on mmWave radar data outperforms state-of-the-art baselines, even for distances beyond the 50 m range.

### 2.2. Fusion-Based Methods

There is a limited number of studies that utilize multiple sensory modalities for 2D object detection in urban driving environments. González et al. [28] employ a mixture-of-experts framework to combine images, depth, and optical flow for 2D pedestrian detection. Enzweiler et al. [29] fuse RGB and depth images at an early stage and train pose-based classifiers for 2D detection. Chen et al. [30] combined camera images with both front and BEV LiDAR perspectives for object detection.

Integrating deep neural networks with radar–vision fusion has gained considerable attention, particularly with the advancements in deep learning. This integration entails leveraging mmWave radar points to capture valuable velocity and depth information. A recent study conducted by Nobis et al. [6] explores the fusion of image and mmWave radar data at the feature level, specifically for object detection in autonomous vehicles. This approach involves extracting features from irregular and sparse mmWave radar point clouds, enabling more comprehensive and accurate object detection. Chadwick et al. [31] demonstrated that integrating radar data can improve performance in challenging scenarios, such as detecting small targets at long distances. John et al. [32] proposed a novel deep-learning-based framework, RVNet, to overcome this limitation. RVNet is a single-shot object detection network that incorporates two input branches and two output branches. Inspired by the YOLO [16] framework, RVNet specifically utilizes two input branches corresponding to the monocular camera and mmWave radar sensor. To effectively fuse the feature maps obtained from the mmWave radar and vision inputs, RVNet employs concatenation neural network layers. However, the presence of the two output branches in RVNet introduces challenges in terms of weighing the loss during model training.

To achieve better fusion of mmWave radar data and vision data for object detection, CRFNet [6] introduced an automatic learning approach based on RetinaNet [33]. It

incorporates a novel training strategy focusing on learning from a specific sensor type. Furthermore, CRFNet introduces a noise filter to enhance the performance of the fusion detection model by effectively filtering out noise present in mmWave radar points. In recent developments, attention mechanisms [34] have demonstrated significant advantages in visual tasks. Huo et al. [2] proposed a novel object detector called SAFF-SSD, which combines self-attention with feature-fusion-based SSD for small object detection. Li et al. [35] proposed an architecture that integrates a feature pyramid attention module to fuse projected mmWave radar data and camera data, expanding the input interface to include mmWave radar projection images and attention modules. Chang et al. [7] proposed a spatial attention fusion (SAF) block to learn the relationship between mmWave radar data and vision data. This method improves detection performance across small, medium, and large scales. In addition, Cheng et al. [8] put forward a resilient radar–vision fusion approach called RISFNet for detecting small objects on water surfaces. RISFNet uses mmWave radar point density maps, which provide spatial distribution details, along with the Doppler velocity and energy of mmWave radar point clouds.

Multiple studies have also integrated LiDAR data and camera images for visual tasks in urban driving environments [36–39]. MVX-Net [36] put forward an early fusion approach that is both straightforward and effective in combining RGB and point cloud modalities. This approach capitalizes on the VoxelNet [24] architecture, which was recently introduced and has demonstrated promising performance in similar tasks. LaserNet++ [39] introduced a technique that integrates image data with LiDAR data, obviating the need for image labels. By incorporating image data, the model's performance can be improved without the requirement of additional labeling efforts. Zhao et al. [40] proposed a novel network architecture that leverages front-view images and frustum point clouds to generate accurate 3D detection results. PointRCNN [41] was specifically developed for 3D object detection using raw point clouds. This method initially generates bottom-up 3D proposals and subsequently refines the proposals in canonical coordinates to obtain the final detection outcomes.

To the best of our knowledge, there is no work that utilizes attention-based networks and fuses spatial–temporal visual-LiDAR-radar data for 2D object detection. In this article, we propose a novel deep fusion network leveraging the strengths of multiple sensors, including LiDAR, mmWave radar, and vision for robust and accurate 2D object detection.

## 3. Methods

In this section, we first present the employed representation formats of different sensor modalities, especially for 3D LiDAR point clouds and 4D mmWave radar point clouds. Subsequently, we introduce the details of the proposed attention-based multi-modal fusion network, LRVFNet.

### 3.1. Representation of Different Modalities

The RGB image is typically represented as a matrix with a size of $W \times H \times C$, where $W$ and $H$ denote the width and height of the image, and $C$ represents the number of RGB color channels. Unlike images, the representations of LiDAR and mmWave radar data are not fixed. LiDAR scanners obtain precise measurements of the shape of the obstacles in the surrounding environments. Existing work usually encodes a 3D LiDAR point cloud into a 3D grid [42] or a front view map [43]. While the 3D grid representation preserves most of the raw information of the point cloud, it usually requires much more complex computation for subsequent feature extraction. Here, we use a more compact representation by projecting a 3D point cloud to BEV as shown in Figure 1.

More specifically, we project each LiDAR or mmWave radar point cloud data into a two-dimensional grid with a resolution of 0.1 m. We normalize the maximum value of the height of all points in each grid and convert it into a gray value using the height information. The generation steps of the BEV map are as follows. We first set the range of

the BEV image as $\{X, Y, Z \mid X \in [0, 40], Y \in [-20, 20], Z \in [-2, 0.5]\}$. Taking a LiDAR point $\mathbf{p} = (x_l, y_l, z_l)$ as an example, we transform it into the camera coordinates by:

$$\begin{bmatrix} x_c & y_c & z_c & 1 \end{bmatrix}^T = \begin{bmatrix} \mathbf{R}_l^c & \mathbf{t}_l^c \\ 0 & 1 \end{bmatrix} \begin{bmatrix} x_l & y_l & z_l & 1 \end{bmatrix}^T, \tag{1}$$

where $\mathbf{R}_l^c$ and $\mathbf{t}_l^c$ represent the rotation and translation matrix from the LiDAR to the camera coordinate systems. $(x_c, y_c, z_c)$ are the point coordinates in the camera coordinate system. We then define the size of a single grid unit according to the set range and the resolution of the BEV image. In our method, we divide the area into multiple small grids with a resolution of 0.1 m on both the $x$-axis and the $y$-axis resulting in a BEV image with the size of $640 \times 640$. After dividing the detection area into a grid, we identify the point set $\mathcal{P}_{i,j} = \{\mathbf{p}\}$ in each cell, where $i, j$ are the respective cell coordinates. Take the height $h_{i,j}$ of the highest point in each cell as the grid value. The conversion method is as follows:

$$h_{i,j} = \max(\mathcal{P}_{i,j}[0\ 0\ 1]^T) \quad . \tag{2}$$

Finally, $h_{i,j}$ is mapped between 0 and 255, converting the height information of the point cloud into the image gray value $g_{i,j}$.

The use of BEV representation offers several advantages. Firstly, it enables more efficient processing compared to point-based methods. Additionally, leveraging mature 2D detection techniques can facilitate the learning process. Moreover, occlusion, a common challenge in distance views and other representations, is alleviated in the BEV. It is worth noting that using BEV representation simplifies and enhances the effectiveness of multimodal fusion strategies. The BEV representation maps three-dimensional point cloud data onto a two-dimensional plane, allowing the data to be presented as a two-dimensional image containing height, width, and channel dimensions, effectively reducing data complexity and improving computational efficiency. At the same time, the data stability of BEV representation is relatively high, which can better adapt to the vehicle's motion states.

### 3.2. Multi-Modal Fusion Network for Object Detection

As shown in Figure 2, our network architecture consists of three modules: backbone, feature fusion, and feature pyramid networks (FPNs). The two-branch backbone architecture is important for extracting features from diverse modalities. By extracting features from the image, LiDAR, and mmWave radar data, our network incorporates valuable information regarding object appearance, context, and spatial relationships among objects in the surrounding environment. The fusion module then plays a vital role in combining features from multiple modalities. To enhance the accuracy and robustness of our network, we employ self-attention blocks that emphasize important spatial and channel-wise features in the fused data. Additionally, a global attention module is utilized to combine multi-scale mmWave radar and image features, effectively capturing rich contextual information and improving the overall performance of the detection task. In the end, we utilize an FPN architecture for object estimation, allowing our model to dynamically incorporate multi-scale features and predict object detection results across different image scales. We provide more detailed descriptions of these modules in the following sections.

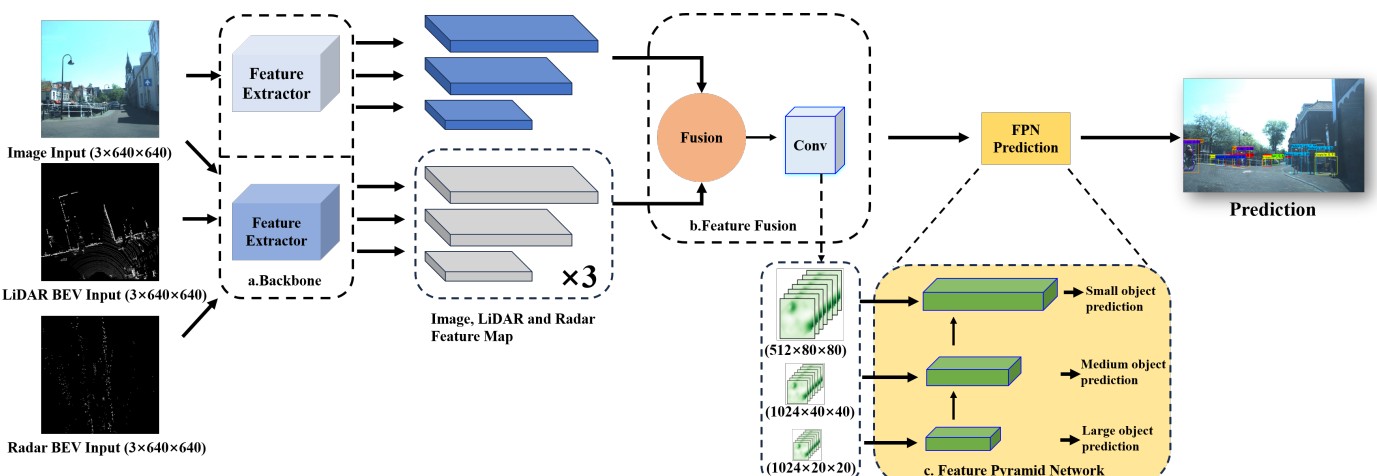

**Figure 2.** A pipeline overview of our proposed LRVFNet, consisting of three modules, the encoder module, feature fusion module, and pyramid object detection module.

### 3.2.1. Two-Branch Backbone Module

Backbone is the main feature extraction module of our LRVFNet, where the input data are first processed to extract features. The main role of the backbone is to extract different features from three different modalities for the next further object reasoning. RGB images and BEV views generated from LiDAR and mmWave radar data are distinctly different regarding their characteristics and information content. RGB images contain more richly detailed visual information than BEV views, primarily presenting a top-down view of the environment. As such, for effective LiDAR-radar and image feature extraction, it is necessary to use different backbone networks for the two modalities. By adopting this approach, the overall efficiency and accuracy of the model can be improved. The backbone network for image feature extraction is illustrated in Figure 3. The choice of the backbone network for LiDAR-radar data processing is a crucial factor in determining the performance of the network. In this regard, we use a lightweight backbone network that is suited for feature extraction from BEV images. The advantage of this approach lies in the ability to extract useful feature information from BEV images with minimal computational cost. Additionally, for feature extraction from BEV, we selected a backbone architecture known as CSPdarknet53 [44], which has been previously demonstrated to be effective in YOLOv4 [45], which extracts image features of varying sizes. Overall, adopting distinct backbone networks for LiDAR-radar and image feature extraction ensures that each modality is processed optimally, leading to improved performance and accuracy of the model.

To increase the receptive field and adapt the algorithm to different-resolution images, we introduce the spatial pyramid pooling–cross stage partial channel (SPPCSPC) module, which obtains different receptive fields through max pooling. As shown in Figure 4, we use four different branches that undergo max pooling with sizes of $\{5, 9, 13, 1\}$. These four different max pooling operations enable the module to handle objects of varying scales by providing four different receptive fields and allowing for distinguishing between large and small targets. For example, in an image containing pedestrians and cars with different scales, it becomes easier to differentiate between small and large targets with the help of our proposed SPPCSPC module.

More specifically, our SPPCSPC first uses the CSP module to initially divide features into two parts. One part undergoes regular processing, while the other part undergoes SPP structure processing. Finally, these two parts are merged together. This approach reduces computation by half, resulting in faster speed, while paradoxically improving accuracy.

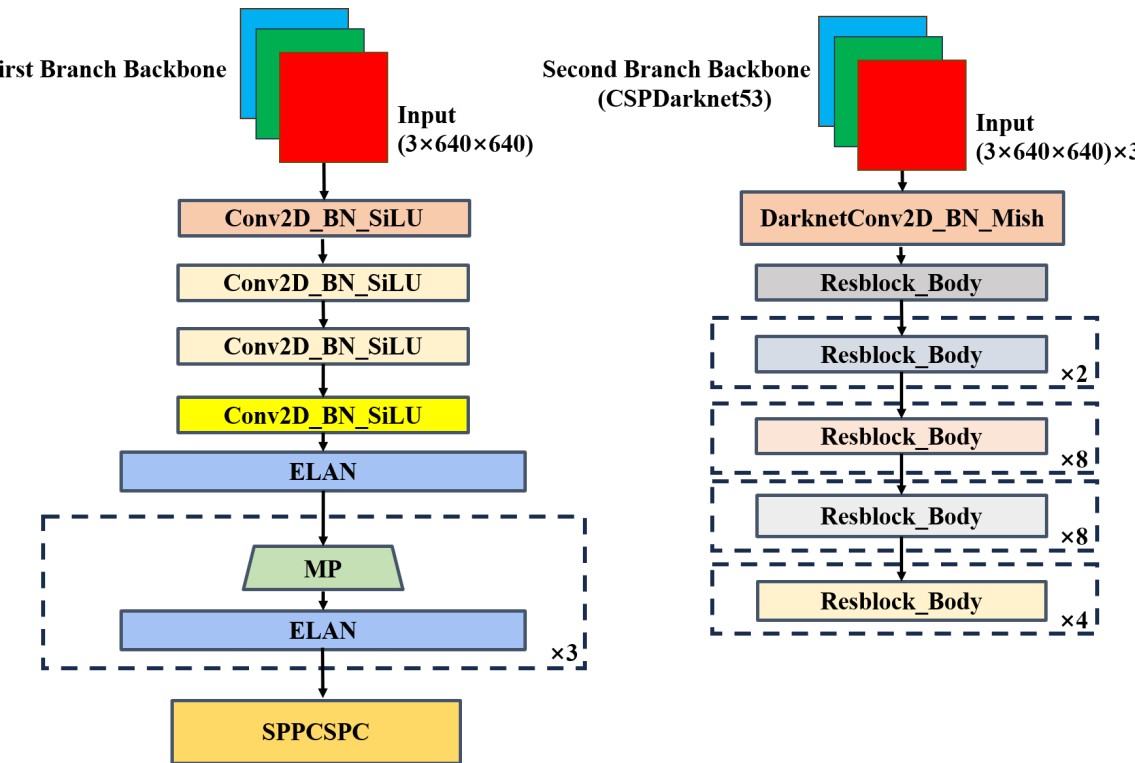

**Figure 3.** The proposed backbone for feature extraction network. The first branch backbone performs multiple convolutions, batch normalization (BN), and SiLU activation on the input, and incorporates max pool (MP) and the advanced efficient layer aggregation networks (ELANs). For the second branch backbone, we adopt CSPDarknet53, which is primarily composed of a series of residual structures.

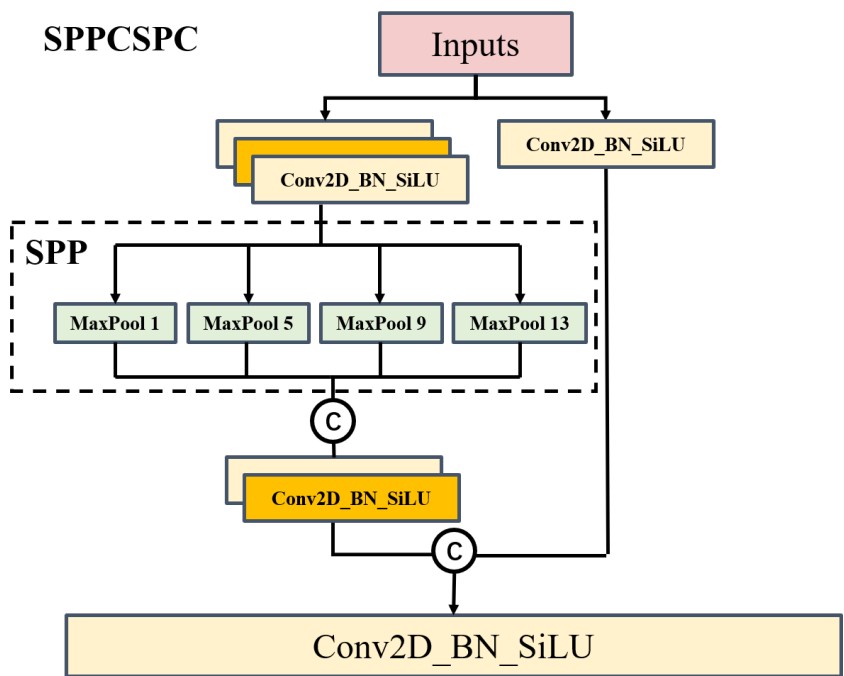

**Figure 4.** The composition of the SPPCSPC module is illustrated in the figure, where the four distinct max pooling operations represent its capability to handle various objects. The CSP module reduces computational load by half, enhancing speed without compromising accuracy.

### 3.2.2. Feature Fusion Module

The fusion module consists of two blocks: the self-attention block and the global attention block. The self-attention block [46] focuses on capturing local dependencies within each modality, while the global attention block aims to capture global dependencies across all modalities. The self-attention block enhances the representation learned within each modality by attending to informative regions, whereas the global attention block [47] facilitates the integration of complementary information across modalities.

The self-attention concept was initially formulated for the purpose of natural language processing and image transformation tasks [46]. A self-attention block functions by allowing individual sensor branches to autonomously adapt themselves, serving as a promising method to regulate information flow and facilitate model adaptation [8]. LiDAR and mmWave radar data encompass valuable 3D information about targets. However, due to measurement noise, especially from mmWave radar data, some points may lead to false object information, introducing errors in the detection results. In such cases, the self-attention module can effectively attenuate noisy points and learn the relationship between LiDAR and mmWave radar points through independent multi-layer perceptron (MLP) blocks as illustrated in Figure 5.

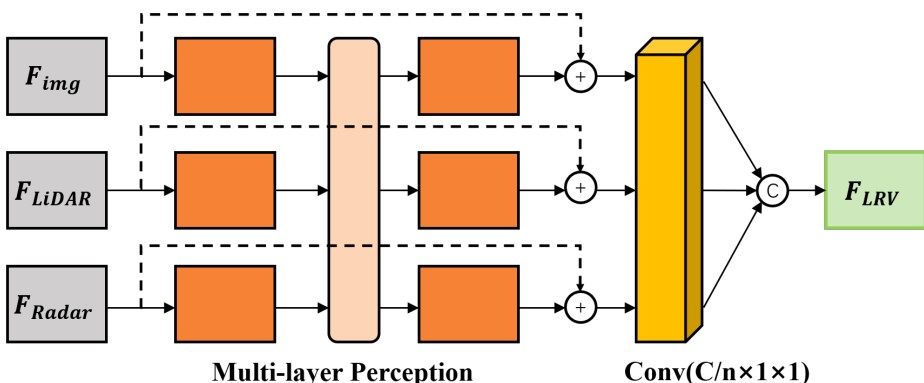

**Figure 5.** The structure of self-attention consists of several independent multi-layer perceptron blocks, where the outputs serve as inputs to the global attention mechanism.

By utilizing multiple independent MLP blocks, the features extracted from images, LiDAR, and mmWave radar data from various frames are individually processed, resulting in the generation of $\mathbf{F}'_{mk} \in \mathbb{R}^{1 \times H \times W}$. Subsequently, a concatenation operation is employed to merge the feature maps of all modalities from different frames, resulting in the creation of a fusion feature denoted as $\mathbf{F}_{LRV}$, formulated as:

$$\mathbf{F}'_{m_k} = Conv(\mathbf{F}_{m_k} + MLP_k(\mathbf{F}_{m_k})) \tag{3}$$

$$\mathbf{F}_{LRV} = Cat(\mathbf{F}'_{m_1}, \mathbf{F}'_{m_2}, \mathbf{F}'_{m_3}) \tag{4}$$

where $\mathbf{F}_{m_k} \in \mathbb{R}^{C \times H \times W}$ denotes the feature map of the $m_k$ modality data; $C, H, W$ denote the channel, height, and width of the feature map, respectively (the values of $C, H, W$ are different under different feature scales); $MLP_k$ is the independent MLP for $\mathbf{F}_{m_k}$; $Conv \in \mathbb{R}^{C/n \times 1 \times 1}$ denotes convolution module to fuse channels; and $Cat(\cdot)$ is a concatenation operation.

By integrating self-attention into the fusion process for target detection, our proposed fusion module effectively combines the complementary strengths of diverse sensor modalities while effectively mitigating the side effects of noisy or irrelevant features. In the context of object detection, it is often necessary to fuse visual data from images with LiDAR and mmWave radar data obtained from sensors. Self-attention provides a mechanism to assign higher importance to specific regions of the image or particular aspects of the LiDAR or radar signal based on their relevance to the detection task at hand.

To further fuse features from different modalities, we then apply a multilayer global attention network for several reasons. Firstly, it is capable of handling complex nonlinear feature spaces with greater ease, which is crucial as real-world environments often exhibit nonlinearity, such as dynamic objects. Additionally, the multilayer global attention network considers all sensor channels simultaneously, which can identify and exploit the complementary behaviors exhibited by each sensor. The structure of global attention is illustrated in Figure 6. This comprehensive utilization of complementary information leads to a more robust fusion model [47], which is particularly important in scenarios where individual sensor measurements may be unreliable or incomplete.

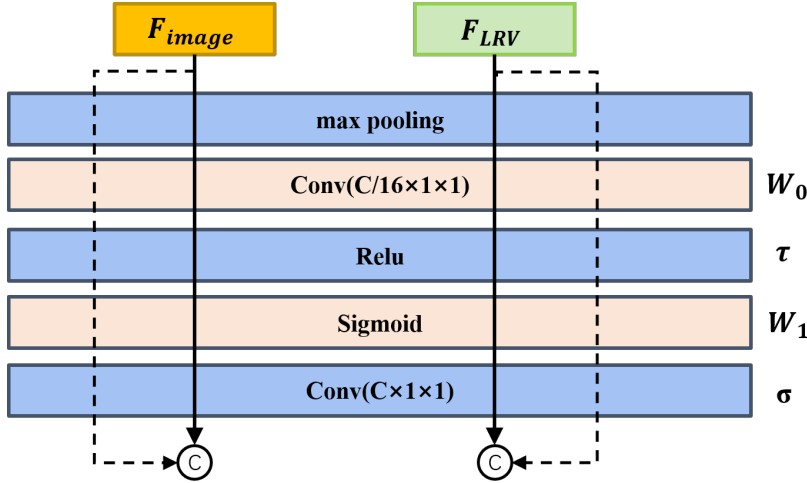

**Figure 6.** The structure of global attention.

As shown in Figure 6, we use a shared *MLP* block to generate deep fusion features $\mathbf{F}_{fusion}$ from image features $\mathbf{F}_{image}$ and three modalities' features $\mathbf{F}_{LRV}$. The global channel attention fusion is computed as:

$$
\begin{aligned}
\mathbf{F}_{fusion} = \sigma(\mathbf{W}_{1\tau}(\mathbf{W}_0(MaxPool(\mathbf{F}_{img}))) + \\
\mathbf{W}_{1\tau}(\mathbf{W}_0(MaxPool(\mathbf{F}_{LRV}))))
\end{aligned}
\tag{5}
$$

where $\sigma$ denotes the sigmoid function and $\tau$ denotes the RELU function. MLP weights $W_0$ and $W_1$ are shared for both image and LiDAR-radar BEV inputs.

By integrating attention mechanisms, the network continuously learns and adjusts its fusion strategy based on the evolving sensor inputs. This adaptive capability allows the fusion model to effectively handle variations in sensor behavior, environmental conditions, and the presence of uncertainties, ensuring reliable and accurate fusion performance.

### 3.2.3. Feature Pyramid Network

A feature pyramid network (FPN) is an enhanced feature extraction network, where the attention-enhanced features are then exploited at different scales for object estimation. It uses the Panet structure [20], which combines upsampling and downsampling to achieve multi-scale feature fusion. As shown in Figure 7, after the fusion of two backbones, we adopt the FPN feature pyramid to improve both accuracy and speed. It replaces the feature extractor in models like Faster R-CNN [13] and generates a higher-quality feature map pyramid. After a series of convolutions, we obtain the feature map. Through upsampling and progressively reducing the size, we preserve the high-level semantic information while increasing the size of the feature map. Then, we use the larger-sized feature map to detect small objects, thus addressing the challenge of detecting small objects.

We adopt the same classifier and regression network as Yolo head [20], in which three effective enhanced feature layers are obtained through Backbone and FPN. Each feature layer contains information about its width, height, and channels. The feature maps can be

understood as a collection of feature points, where each point is associated with three prior bounding boxes, and each bounding box holds features that are relevant to a specific object class. The Yolo head is in charge of determining if an object is present for each prior at each feature point. Eventually, the classification and regression processes are integrated through a $1x1$ convolution to yield the ultimate outcome of object detection.

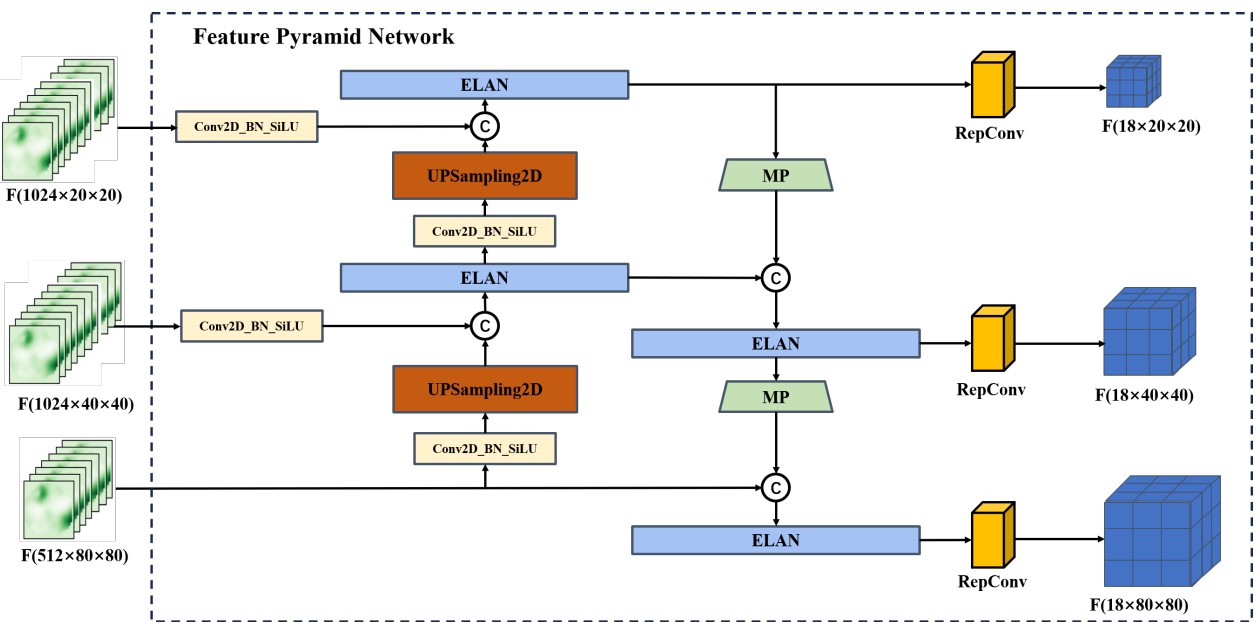

**Figure 7.** Architecture of our feature pyramid network is used for enhanced feature extraction.

### 3.3. Loss Functions

The loss function $L$ of our network consists of three components: the regression part $L_{Reg}$, the object part $L_{Obj}$, and the classification part $L_{Cls}$. $L_{Reg}$ is responsible for determining the regression parameters of the anchor points. $L_{Obj}$ determines whether the anchor points contain objects, while $L_{Cls}$ identifies the specific category of objects contained within the anchor points.

$$L = L_{Reg} + L_{Obj} + L_{Cls} \tag{6}$$

The regression part $L_{Reg}$ is formulated based on the object detection IOU as:

$$L_{Reg} = 1 - IoU(A, B) + \rho^2(A_{ctr}, B_{ctr})/c^2 + \alpha v \ , \ IoU(A, B) = \frac{A \cap B}{A \cup B} \tag{7}$$

where $A$ represents the predicted bounding box and $B$ represents the ground truth bounding box. $A_{ctr}$ denotes the coordinates of the center point of the predicted bounding box, while $B_{ctr}$ denotes the coordinates of the center point of the ground truth bounding box. $\rho$ represents the calculation of the Euclidean distance. $c$ denotes the diagonal length of the minimum bounding box that encloses A and B. $v$ is used to measure the consistency of aspect ratios. $\alpha$ is a weighting parameter, while $w$ and $h$ denote the width and height of the ground truth box, and $w^{gt}$ and $h^{gt}$ denote the width and height of the predicted box.

For $L_{Obj}$, we modify the cross-entropy loss function by incorporating the IoU metric between the predicted bounding boxes and the ground truth. When the IoU exceeds a predefined threshold, the sample is classified as a positive sample, indicating the presence of an object. Otherwise, it is classified as a negative sample, indicating the absence of an object.

$$L_{Obj} = -\frac{1}{N} \sum_{i=1}^{N} [IoU_i \log(p_i) + (1 - IoU_i \log(1 - p_i))] \quad . \tag{8}$$

$L_{Cls}$ is computed based on the real category of the boxes and the predicted category results of the anchor boxes, formed as:

$$L_{Cls} = -\frac{1}{N}\sum_{i=1}^{N}[IoU_i\log(p_i) + (1 - IoU_i)\log(1 - p_i))], \qquad (9)$$

where $N$ is the total number of samples, $y_i$ is the ground truth label for sample $i$ (1 for positive samples and 0 for negative samples), $p_i$ is the predicted probability for sample $i$, and $-\frac{1}{N}$ is a normalization factor to average the loss over all samples.

## 4. Results

### 4.1. Dataset

We evaluate our LRVFNet on the public VOD dataset [9] and Flow dataset [10] collected in real-world urban environments. The VOD dataset [9] consists of a 64-beam LiDAR, binocular cameras, and a 4D mmWave radar, containing a total of 8693 synchronized and calibrated frames in a challenging urban traffic environment. The dataset provides 123,106 3D bounding boxes, including 26,587 pedestrians, 10,800 cyclists, and 26,949 cars. Flow dataset [10] collected $1280 \times 720$ RGB images at 15 Hz. The IMU recorded pose information at a frequency of 10 Hz, while the mmWave radar employs a 77 Ghz FMCW radar. The frame rate of the mmWave radar is also 10 Hz. To ensure synchronization, data from different sensors are aligned using recorded timestamps.

### 4.2. Implementation

We split each dataset into the training set and the test set in a ratio of 4:1 following [8]. For enriched training, we employ multi-scale data augmentation techniques. These include image resizing, where the training images and BEVs are resized to different scales; image placing, which involves randomly positioning training images within a larger canvas; and image left–right flipping, which artificially increases the diversity of the training data. Such data augmentation can effectively improve the generalization capabilities of deep learning models.

In the training, we use the model CSPDarknet53 pretrained on VOC datasets [48] for the fusion branch of the image backbone. Our implementation is based on PyTorch and trained on four Nvidia Geforce RTX 4090 GPUs with an initial learning rate set to be $1 \times 10^{-3}$ and batch size set to be 4. During the testing process, our LRVFNet model achieves an average running speed of approximately 44.25 frames per second (FPS) on a device equipped with an Nvidia Geforce RTX 4090 GPU. This indicates that our model can meet the real-time requirements for object detection in autonomous driving.

### 4.3. Evaluation Metrics

To evaluate the prediction result of the model, the 2D bounding box average precision (AP) under different IoU thresholds is employed as the evaluation metric. This metric evaluates the model's precision and recall performance simultaneously. A higher AP value indicates better prediction precision of the bounding box and a smaller missing detection rate of the objects. To identify multi-category objects, mean average precision (mAP) is typically employed, where mAP is calculated as the average of individual category APs. The true positive (TP) prediction boxes overlap with the ground truth objects over the IoU threshold, while false positive (FP) objects overlap with ground truth objects less than the IoU threshold. The missing objects are treated as false negatives (FNs). The precision $P$ and recall $R$ are calculated as:

$$P = \frac{TP}{TP + FP} \ , \ R = \frac{TP}{TP + FN} \qquad (10)$$

.

AP value is then calculated by taking the mean of all distinct precision values corresponding to different recall points.

$$AP = \sum_{i=1}^{n-1}(R_{i+1} - R_i)P(R_{i+1}) \ , \ mAP = \frac{\sum_{i=1}^{K} AP_i}{K},  \tag{11}$$

where *K* represents the summation of all categories in the object detection task.

$AP^{35}$ denotes AP at IoU = 0.35. $AP^{50}$ denotes AP at IoU = 0.5. $AP^{35}$ and $AP^{50}$ are important metrics for evaluating the performance of object detection algorithms, as they can assess the accuracy of the algorithms at different IoU thresholds. In autonomous driving tasks, the precise detection of surrounding obstacles, vehicles, and pedestrians is crucial. However, these targets vary in size and shape, making it necessary to evaluate algorithm performance using multiple IoU thresholds to obtain a comprehensive assessment. Choosing $AP^{35}$ and $AP^{50}$ as evaluation metrics helps distinguish the performance of object detection algorithms. Higher $AP^{35}$ and $AP^{50}$ values indicate that the algorithm can detect targets more accurately and provide more precise position information. Accuracy and precision are crucial in autonomous driving tasks; therefore, selecting these two metrics facilitates evaluating whether the algorithm meets the requirements of autonomous driving tasks.

### 4.4. Evaluation Results

To evaluate the performance of the proposed multimodal fusion involving LiDAR, mmWave radar, and visual sensors, we conducted experiments using the VOD dataset [9]. We compared our proposed LRVFNet with two widely used RGB-based methods [20,45] and our LRVFNet under modal degradation (without LiDAR) against RISFNet [8]. For a fair comparison, we also compared our LRVFNet using three modalities with our reimplemented RISFNet using three modalities. The experimental results are presented in Table 1. Our method achieved the highest AP values compared to the state-of-the-art visual detection method, Yolov7 [20], with improvements of 5.85% and 4.56% at IoU thresholds of 0.5 and 0.35, respectively. Furthermore, our model outperformed the degraded LiDAR model by 1.47% and 1.07% in AP at the respective thresholds, demonstrating the benefits of using the LiDAR input. Moreover, even under model degradation, our LRVFNet still exhibited robust performance, showing its resilience.

**Table 1.** Results on the VOD dataset using our method and other multimodal fusion methods.

| Modality | Method | $AP^{35}$ | $AP^{50}$ |
|---|---|---|---|
| Image | Yolov4 [45] | 31.06 | 29.78 |
|  | Yolov7 [20] | 83.74 | 80.65 |
| Image + Radar | RISFNet [8] | 66.78 | 45.55 |
|  | LRVFNet (Ours) | 88.30 | 85.03 |
| Image+LiDAR+Radar | RISFNet + LiDAR | 76.72 | 54.23 |
|  | LRVFNet (Ours) | **89.37** | **86.50** |

To verify the improvement in detection accuracy using our proposed fusion method, we compared our method with state-of-the-art methods using a single modality and two modalities on the FLOW dataset [10]. The dataset contains continuous sequences of synchronized images and mmWave radar data with accurate annotations. As shown in Table 2, we compared our LRVFNet using only image–radar data with four RGB-based methods and three fusion methods. The training and testing sets used in all the baseline methods are the same as ours. The results show that our method achieves improved AP compared to the visual-only detection method, FCOS [49], with an increase of 19.9% and 22.41% for IoU thresholds of 0.5 and 0.35, respectively. This improvement is attributed to the enriched sensor information provided by the fusion of multiple modalities and the efficient fusion strategy employed. Furthermore, our method outperforms RISFNet [8],

with a performance improvement of 3.37% and 1.07% for IoU thresholds of 0.5 and 0.35, respectively. This is due to our network's superior ability to detect small and occluded objects. Additionally, the results demonstrate that our model maintains high robustness and achieves optimal performance even under modality degradation.

**Table 2.** Result on Flow dataset [10] using our method and other multimodal fusion methods.

| Modality | Method | $AP^{35}$ | $AP^{50}$ |
| --- | --- | --- | --- |
| Image | Faster-RCNN [13] | 77.35 | 57.58 |
| | Yolov4 [45] | 78.46 | 57.04 |
| | EfficientDet [15] | 78.62 | 58.52 |
| | FCOS [49] | 68.71 | 58.56 |
| Image + Radar | CRF-Net [6] | 79.63 | 57.74 |
| | Li et al. [35] | 85.28 | 64.64 |
| | RISFNet [8] | 90.05 | 75.09 |
| | LRVFNet (Ours) | **91.12** | **78.46** |

To gain more insights into the performance of various modalities in object detection, we provide qualitative results in Figure 8. It is clear that integrating LiDAR, mmWave radar, and visual sensors yielded superior outcomes compared to YOLOv4, which relies only on image detection, and RISFNet, which relies on image and mmWave radar data. The information fusion of the three sensors can offer valuable distance information, proving particularly advantageous for detecting small targets at extended distances. As can be seen from the first row in the figure, our method can also be successfully identified when the object in the image information is occluded. LiDAR and mmWave radar data actively contribute to the fusion process, thereby enhancing overall detection accuracy.

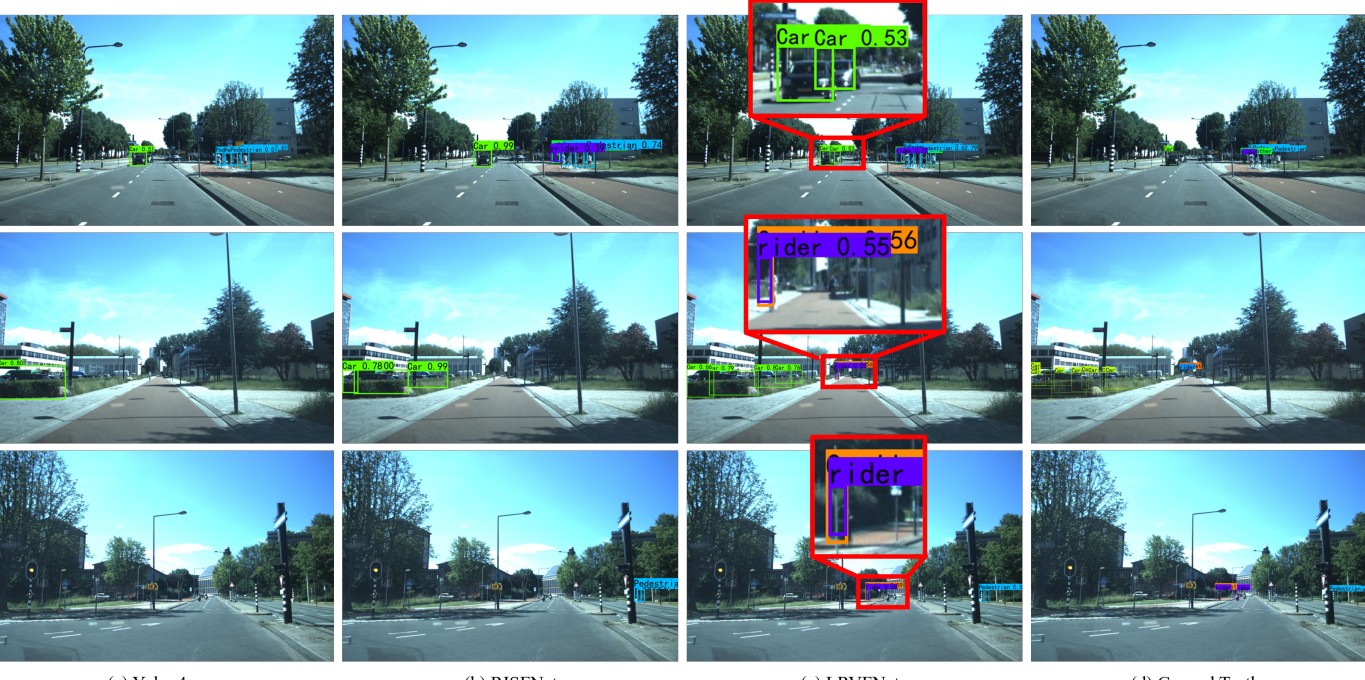

(a) Yolov4  (b) RISFNet  (c) LRVFNet  (d) Ground Truth

**Figure 8.** Qualitative results on the VOD dataset. As depicted in the figure, we show the results of different methods, (**a**) Yolov4, (**b**) RISFNet, and (**c**) LRVFNet (ours). We also provide object detection ground truth (**d**). It can be observed that LRVFNet demonstrates excellent capability in 2D object detection and performs well in recognizing objects with long distances and large occlusions.

In summary, our LRVFNet approach, leveraging the fusion of LiDAR, mmWave radar, and visual sensors with the incorporation of attention mechanisms, demonstrated superior

performance in 2D object detection. The comprehensive experimental analysis conducted on the Flow and VOD datasets confirmed the effectiveness and robustness of our proposed method in autonomous driving scenarios.

### 4.5. Ablation Study

To validate the effectiveness of each devised module in our LRVFNet, we conducted a comprehensive ablation study. The results obtained, as presented in Table 3, clearly demonstrate a significant decrease in performance measures for both $AP^{35}$ and $AP^{50}$ tests. We show the importance of the self-attention module in capturing and leveraging contextual dependencies within the input data. Similarly, when we examined the model's performance in the absence of the global attention module, we observed a consistent decline in performance. This further accentuates the critical role played by the global attention module in capturing long-range dependencies and facilitating a holistic understanding of the scene.

**Table 3.** Results of ablation study on model architecture.

| Ablation Ways | $AP^{35}$ | $AP^{50}$ |
|---|---|---|
| only use one backbone | 82.28 | 79.15 |
| no use self attention | 87.17 | 82.32 |
| no use global attention | 86.74 | 80.07 |
| 1 scans Radar | 89.28 | 85.79 |
| 3 scans Radar | **89.37** | **86.50** |

The performance of the model without the self-attention block decreased by 4.18% and 2.2% respectively. Similarly, the model without the global attention block exhibited performance degradation of 6.43% and 2.63% respectively. These findings suggest that the introduced attention mechanisms effectively fuse information from LiDAR, mmWave radar, and vision, enabling efficient object detection. Considering the sparsity in the mmWave radar point cloud, we conducted ablation experiments by varying the number of frames used for the mmWave radar point cloud. The results indicate that using three frames of mmWave radar point clouds improves performance by 0.71% and 0.09% compared to using a single frame, demonstrating that the spatial–temporal coherence of mmWave radar point clouds provides richer information.

As illustrated in the qualitative results in Figure 9, the fusion of image and mmWave radar notably enhances object detection effectiveness, enabling the detection of a larger number of objects than purely image inputs. In the first set of results, (b) demonstrates the detection of pedestrians at long range through the fusion of mmWave radar, with higher confidence in the recognition of nearby targets. Additionally, the fusion of LiDAR data further elevates detection performance, as shown in (c), where not only pedestrians are successfully detected but also a rider at a greater distance. In the second set, two pedestrians are successfully detected at a long range, whereas the fusion of mmWave radar and visual sensors only manages to detect one pedestrian, thus demonstrating the superior accuracy of the three-modal fusion model incorporating LiDAR information over visual and two-modal fusion.

Building on these insights, our proposed approach leverages attention modules to effectively fuse information from multiple sensors, thereby exploiting the distinctive characteristics of both LiDAR and mmWave radar. By doing so, our method significantly enhances the performance of 2D object detection on autonomous driving datasets.

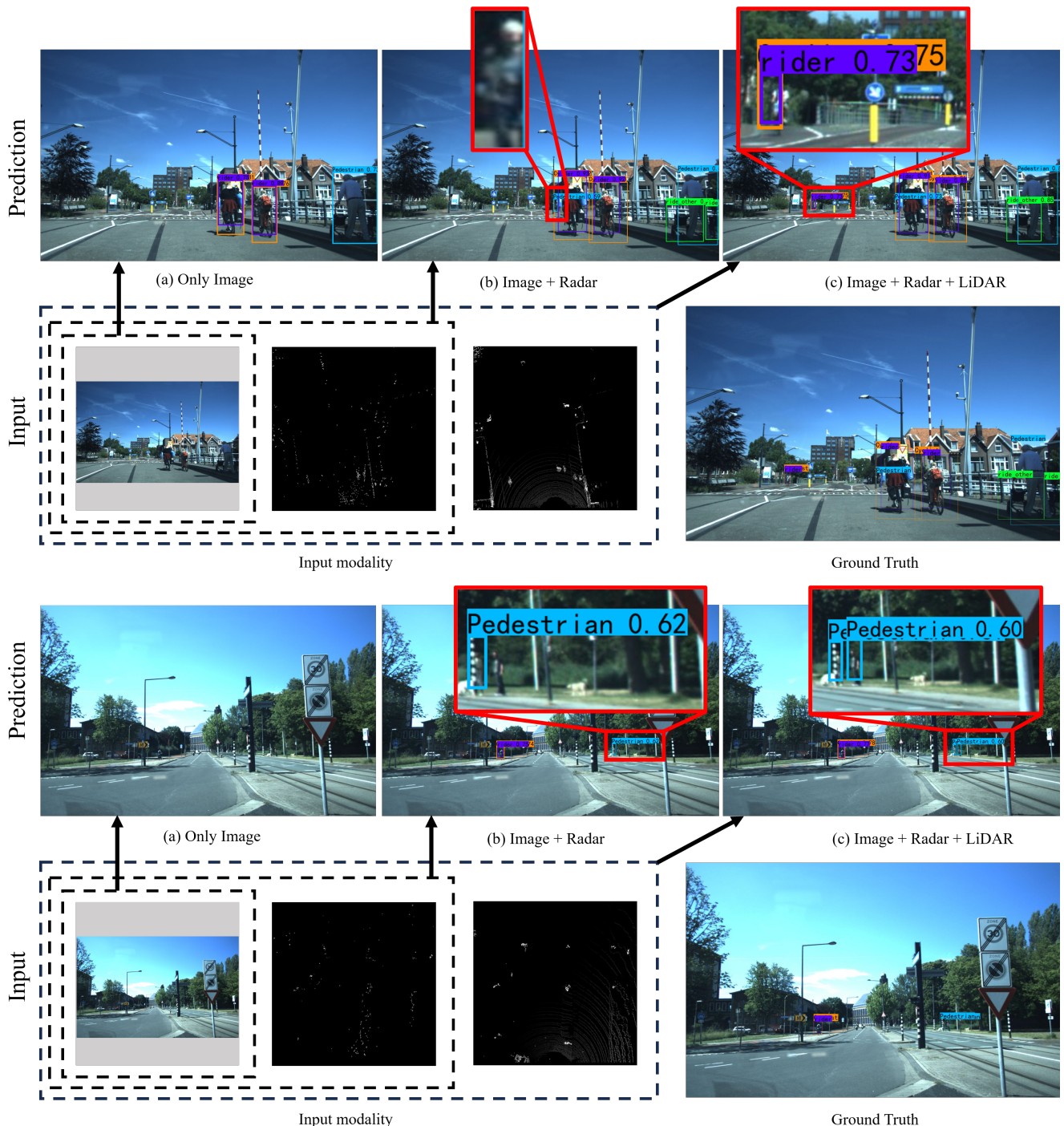

**Figure 9.** Experimental results under different modal inputs. As depicted in the figure, the first row represents the detection results under different modal inputs, (**a**) using only images, (**b**) image–radar, and (**c**) fusion of three modalities. The second row indicates the different inputs. We also attach the object detection ground truth. The results demonstrate that our approach successfully integrates multimodal information, leading to a significant improvement in detection performance.

## 5. Conclusions

In this article, we proposed a novel deep fusion algorithm exploiting LiDAR, mmWave radar, and camera data, referred to as LRVFNet, to achieve robust and accurate 2D object detection in urban driving environments. LRVFNet utilizes the LiDAR and mmWave radar input format in the form of BEV and incorporates attention mechanisms for efficient fusion. It furthermore exploits spatial–temporal information from multiple modalities, maintaining

robustness when any modality degrades. We evaluated our proposed LRVFNet on the VOD dataset and Flow dataset. Both the quantitative and qualitative results demonstrate that LRVFNet outperforms state-of-the-art existing 2D object detection methods.

**Author Contributions:** Conceptualization, Y.X. and X.C.; methodology, Y.X. and Y.L.; software, Y.X.; validation, K.L.; resources, H.L.; writing—review & editing, H.L., X.C. and Y.C.; supervision, X.C.; project administration, H.L. and X.C.; funding acquisition, H.L. All authors have read and agreed to the published version of the manuscript.

**Funding:** This work was supported in part by the National Science Foundation of China under Grant U191320, as well as Major Project of Natural Science Foundation of Hunan Province under Grant 2021JC0004.

**Data Availability Statement:** The data presented in this study are available on request from the corresponding author.

**Conflicts of Interest:** The authors declare no conflict of interest.

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
