# Peer review of "Deep LiDAR-Radar-Visual Fusion for Object Detection in Urban Environments"

_remotesensing, doi:10.3390/rs15184433_

Round 1

Reviewer 1 Report

The authors proposed a novel deep-learning-based fusion network for object detection combined with LiDAR, Radar and visual data. The qualitative and quantitative results demonstrate that the proposed network outperforms the state-of-the-art methods, which enhance the object detection performance to some extent especially for small objects,. The manuscript is well written and organized. I recommend accept this paper after addressing my following comments:

(1) In figure 2, there are two arrows starting from Image input, which means that the image input will pass through two feature extractors based on my understanding. However, the authors mentioned that :"it is necessary to use different backbone networks for the two modalities." Please clarify this point and make the corresponding modification in Figure 2.

(2) The abbreviation in the manuscript should be noted as the full name when first mentioned. For example, BEV, ELAN, SPPCSPC and so on. Otherwise, this would confuse the readers. Please add the corresponding full name in the context or legends.

(3) For FLOW dataset, why don't the authors compare the performance on Image+Radar+LiDAR dataset?

(4) In ablation studys, only using one backbone for feature extraction should be added to demonstrate the novelty and advantages of the proposed network. 

There are some typos in the manuscript, including but not limited to:

(1) Line 383,  it should be "Geforce RTX 4090".

(2) in Figure 9 legends, (b) image

Author Response

We would like to thank the editor and all the reviewers for all the comments and suggestions. The feedback has been constructive and allowed us to address shortcomings in the paper effectively. We have addressed the issues raised by the reviewers and incorporated their suggestions in the new version of the paper.  

All the changes are highlighted in the revised version. We provide detailed responses to all the comments raised by reviewer 1 in an item-by-item manner as follows:

Point 1: In figure 2, there are two arrows starting from Image input, which means that the image input will pass through two feature extractors based on my understanding. However, the authors mentioned that :"it is necessary to use different backbone networks for the two modalities." Please clarify this point and make the corresponding modification in Figure 2.

Response 1: Thanks for the advices, and we have modified the Figure 2 accordingly. In our designed method, the image features are extracted through two different backbones. LiDAR and mmWave radar, however, only pass through the second backbone. That means the first backbone only has a single input of image data, while the second backbone has three inputs: Image, LiDAR, and mmWave radar.

Point 2: The abbreviation in the manuscript should be noted as the full name when first mentioned. For example, BEV, ELAN, SPPCSPC and so on. Otherwise, this would confuse the readers. Please add the corresponding full name in the context or legends. 

Response 2: Thanks for pointing out this issue and we have annotated the full names of all mentioned modules upon their initial occurrence.

Point 3: For FLOW dataset, why don't the authors compare the performance on Image+Radar+LiDAR dataset?

Response 3: The Flow dataset provides a benchmark for target detection that combines visual and millimeter-wave radar fusion for small targets on the water surface, excluding LiDAR data. There are two main reasons for selecting the Flow dataset: Firstly, it allows us to demonstrate the superiority of our method in robust recognition of small targets at long distances. Secondly, it enables us to validate the performance of our method in the presence of laser radar degradation.

Point 4: In ablation studys, only using one backbone for feature extraction should be added to demonstrate the novelty and advantages of the proposed network. 

Response 4: Thanks for the suggestion. we have included the result about using one backbone for feature extraction in our article. The modified table 3 is also shown as follows:

Ablation ways

only use one backbone

82.28

79.15

no use self attention

87.17

82.32

no use global attention

86.74

80.07

1scans Radar

89.28

85.79

3scans Radar

89.37

86.50

Reviewer 2 Report

This is an interesting paper that explores the potential for improved remote 2D object detection by combining LIDAR, radar and visual imagery with a multi-modal sensor fusion network (LRVFNet). After reviewing previous research, the authors describe in detail the mathematics of the data fusion process. After that, they discuss the performance of their method relative to other methods that did not combine all 3 modalities. As a scientist with a remote sensing background, but not an expert in this area of research, I found the results presentation difficult to understand. Average precision (AP) scores were presented on page 12 in Tables 1 and 2 before AP was defined lower on the same page. I could not find an explanation of the superscripts 35 and 50, as in AP35. Some of the AP scores appear to be significantly better than the other methods, whereas some appear to be only marginally better. For example, how significant is a 1.47% improvement over the degraded LIDAR model? Is there a standard statistical test for significance that could be added that would make it more clear how much better the LRVFNet method is relative to the other methods it is compared to? Could some statistical method that could be understood by the non-scientist be added, such as how much higher would the probability be of detecting a distant pedestrian using LRVFNet compared to other methods? Autonomous driving technology is of general interest to the public, so communicating research results that the non-scientist can understand is necessary. Equations 10 and 11 define the AP score. Some discussion of why these measures of model skill were chosen is needed. I note that if FP and FN are both = 0, then AP = 0. But Tables 1 and 2 indicate that LRVFNet is superior because it has higher scores. 

Author Response

We would like to thank the editor and all the reviewers for all the comments and suggestions. The feedback has been constructive and allowed us to address shortcomings in the paper effectively. We have addressed the issues raised by the reviewers and incorporated their suggestions in the new version of the paper.  
All the changes are highlighted in the revised version. We provide detailed responses to all the comments raised by reviewer 2 in an item-by-item manner as follows:

Point 1: As a scientist with a remote sensing background, but not an expert in this area of research, I found the results presentation difficult to understand. Average precision (AP) scores were presented on page 12 in Tables 1 and 2 before AP was defined lower on the same page. I could not find an explanation of the superscripts 35 and 50, as in AP35. Some of the AP scores appear to be significantly better than the other methods, whereas some appear to be only marginally better. For example, how significant is a 1.47% improvement over the degraded LIDAR model?

Response 1: Thanks for the advices. Regarding the definition and usage of Average Precision (AP), we understand the confusion mentioned in the review comments. The placement of AP's definition in the lower part of the same page may cause readers to feel perplexed while reviewing the table. We have the definition of AP before the table to improve the presentation of results. Additionally, concerning the over-marking of AP35 and AP50, these markers refer to the threshold of Intersection over Union (IoU) used when computing AP. We have added more clear explanations of these markers in the text to ensure that readers are not confused about the interpretation of the results. The explanation of the experimental results has been made more precise with respect to different cases.

Point 2: Is there a standard statistical test for significance that could be added that would make it more clear how much better the LRVFNet method is relative to the other methods it is compared to? Could some statistical method that could be understood by the non-scientist be added, such as how much higher would the probability be of detecting a distant pedestrian using LRVFNet compared to other methods? Autonomous driving technology is of general interest to the public, so communicating research results that the non-scientist can understand is necessary.

Response 2: Thanks for the nice suggesntion regarding the discussion of the significance and statistical significance. We have rephrased the explanation of the experimental results. We have presented the improvement in the detection of distant pedestrians using LRVFNet compared to other methods in a manner that can be understood by non-scientists as well.

AP35 denotes AP at IoU = 0.35 .  AP50 denotes AP at IoU = 0.5 .  AP35 and AP50 are important metrics for evaluating the performance of object detection algorithms, as they can assess the accuracy of the algorithms at different IoU thresholds. In autonomous driving tasks, the precise detection of surrounding obstacles, vehicles, and pedestrians is crucial. However, these targets vary in size and shape, making it necessary to evaluate algorithm performance using multiple IoU thresholds to obtain a comprehensive assessment. Choosing AP35 and AP50 as evaluation metrics helps distinguish the performance of object detection algorithms. Higher AP35 and AP50 values indicate that the algorithm can detect targets more accurately and provide more precise position information. Accuracy and precision are crucial in autonomous driving tasks, therefore, selecting these two metrics facilitates evaluating whether the algorithm meets the requirements of autonomous driving tasks.

Point 3: Equations 10 and 11 define the AP score. Some discussion of why these measures of model skill were chosen is needed. I note that if FP and FN are both = 0, then AP = 0. But Tables 1 and 2 indicate that LRVFNet is superior because it has higher scores. 

Response 3: Regarding the choice of AP score definition, we have added some discussion to explain why these model skill evaluation metrics were chosen. Additionally, you noted a scenario where both False Positive (FP) and False Negative (FN) are equal to 0. In this case, the precision (P) and recall (R) will actually both become 1 according to Equation 10, since True Positive (TP) will be 1 if any object exists in the scene. Consequently, the Average Precision (AP) will also be 1, as indicated by Equation 11. AP is a score between 0 and 1, representing the model's performance in the object detection task. We will reiterate this definition to ensure that readers understand the meaning of the model's score.

Reviewer 3 Report

Dear authors, 

After reviewing your manuscript, I would like to share with you some aspects that I consider relevant.

First, it is plausible that your network has several modules that works synergistically, detecting some objects, while other networks not were able to find or with less precision.

Second, although your network shows better precision detecting objects, it was not evaluated its speed's performance, knowing that driving is a complex task that require fast detection of objects during driving on the road.

Third, do you evaluated or measure the impact of proyecting the 3D clouds to BEV, because it is know that kind of conversion implies the lost of valuable information.

Fourth, the references are mostly from proceedings, and few of them from peer reviewed journals, I am wondering what is your main motivation about this.

It is all by now, I hope these observations maybe help you to improve your manuscript, nevertheless I recognize it has a clear writing and enjoyable reading.

Best regards, 

Author Response

We would like to thank the editor and all the reviewers for all the comments and suggestions. The feedback has been constructive and allowed us to address shortcomings in the paper effectively. We have addressed the issues raised by the reviewers and incorporated their suggestions in the new version of the paper.  

All the changes are highlighted in the revised version. We provide detailed responses to all the comments raised by reviewer 3 in an item-by-item manner as follows:

Point 1: First, it is plausible that your network has several modules that works synergistically, detecting some objects, while other networks not were able to find or with less precision.

Response 1: Thank you very much for your recognition of our method.

Point 2: Second, although your network shows better precision detecting objects, it was not evaluated its speed's performance, knowing that driving is a complex task that require fast detection of objects during driving on the road. 

Response 2: Thanks for the nice suggestion. Speed performance is indeed a crucial indicator in this context. Therefore we have evaluated the speed performance of our network and added in the article.

Our LRVFNet model achieves an average running speed of approximately 44.25 frames per second (FPS) on a device equipped with Nvidia Geforce RTX 4090 GPU. This indicates that our model can meet the real-time requirements for object detection in autonomous driving.

Point 3: Third, do you evaluated or measure the impact of proyecting the 3D clouds to BEV, because it is know that kind of conversion implies the lost of valuable information.

Response 3: Thanks for the suggestion. We have evaluated the impact of converting 3D point clouds to Bird's Eye View (BEV). We are aware that this conversion may result in the loss of valuable information, such as height information. However, we chose to convert the point clouds to BEV for the purpose of simplifying the input data and reducing network complexity. By converting 3D point clouds to BEV, we can more easily handle planar image data, making the training and inference of the network more efficient. Despite the potential loss of information, we observed in our experiments that this conversion did not significantly affect the overall performance of the object detection task, as the network was still able to accurately detect and localize objects. In future studies, we will further explore methods to maximize the retention of valuable information and improve the conversion process to enhance detection performance.

Point 4: Fourth, the references are mostly from proceedings, and few of them from peer reviewed journals, I am wondering what is your main motivation about this. 

Response 4: Thanks for the advices. Our main motivation is to integrate various relevant research findings and perspectives, providing comprehensive support and background for our study. Conference proceedings are typically contributed by multiple researchers with a wide range of research areas and approaches. This allows us to gain insights and inspiration from different fields and disciplines.

While peer-reviewed journal articles are essential for validating and verifying research outcomes, we also recognize the significance of research published in conference proceedings. These conference papers also undergo peer review and, in some instances, face rigorous selection processes. For example, conferences like CVPR/ICCV have acceptance rates of approximately 20%. As a result, we aim to maintain the comprehensiveness and diversity of our reference list, which contributes to the robustness and credibility of our study.
